# CONTINUAL REINFORCEMENT LEARNING BY REWEIGHTING BELLMAN TARGETS

## ABSTRACT

One major obstacle to the general AI agent is the inability to solve new problems without forgetting previously acquired knowledge. This deficiency is highly linked to the fact that most reinforcement learning (RL) methods are based upon the key assumption that the environment transition dynamics and reward functions are fixed. In this paper, we study the continual RL setting by proposing a general analysis framework of catastrophic forgetting in value-based RL based on the defined MDP difference. Within this theoretical framework, we first show that without incorporating any strategies, the Finetune algorithm, one commonly used baseline regarded as the lower bound a continual RL algorithm can achieve, suffers from complete catastrophic forgetting. Moreover, the sequential multi-task RL algorithm, normally viewed as one soft upper bound baseline, can lead to an optimal action-state value function estimator at the cost of almost intractable computation cost in an online alternating algorithm. Motivated by these results, a practical continual RL algorithm is proposed by reweighting the historical and current Bellman targets to trade-off between these lower and upper-bound approaches. We conduct rigorous experiments in the tabular setting to demonstrate our analytical results, suggesting the massive potential of our proposed algorithm in real continual RL scenarios.

## 1 INTRODUCTION

It is a major challenge to develop general artificial intelligent agents that can continually learn new tasks while maintaining the knowledge they previously obtained in the historical tasks. This research issue also referred to *continual reinforcement learning (RL)* (Khetarpal et al., 2022), has gained increasing attention in recent years, and efforts to solve it have also grown substantially (Barreto et al., 2020; Kessler et al., 2022; Kaplanis et al., 2019; Caccia et al., 2022; Kaplanis et al., 2018; Gaya et al., 2023; Yang et al., 2023; Wolczyk et al., 2022). Developing effective continual RL algorithms is crucial yet challenging as most successful reinforcement learning algorithms are designed for one Markov decision processes (MDP) (Sutton & Barto, 2018), where they assume the underlying MDP is stationary with a fixed reward function and transition dynamics. However, this assumption not only tends to be violated in practical problems (Chandak et al., 2020), but also limits the generality of RL algorithms to adaptively solve new problems toward human-level agents.

When addressing continual learning problems (Wixted, 2004), it suggests that, unlike the human brain, deep neural networks are prone to *catastrophic forgetting* issue (French, 1999; McCloskey & Cohen, 1989), where deep nets or agents can quickly perform poorly on the previous tasks when they are sequentially trained on a series of new tasks. Despite the promising progress of continual RL to mitigate catastrophic forgetting, we still have a poor understanding of how to clearly define and solve this problem fundamentally. Recently, a conceptual basis of continual RL was provided in (Abel et al., 2023) to formalize the notion of "agents can never stop learning", however, there still exists an explicit gap between their formalism and the practical algorithm design as well as fundamental properties of continual learning, such as catastrophic forgetting, plasticity and stability.

In this paper, we study the foundations of continual RL by proposing a general theoretical analysis framework in this context. Specifically, we leverage the difference of optimal action-state value functions in the respective MDPs to define the MDPs difference, based on which the catastrophic forgetting is explicitly characterized. Within this analytical framework in continual RL, we further

investigate two typical continual RL baselines, i.e., the Finetune and sequentially multi-task learning algorithms, which are commonly viewed as the lower and upper bound approaches, respectively. Concretely, we reveal that the Finetune algorithm suffers from complete catastrophic forgetting, while the sequential multi-task learning is capable of learning an optimal continual Q function estimator at the cost of an almost intractable computational cost in an online alternating algorithm. These results motivate us to propose a practical continual RL algorithm via reweighting previous and current Bellman targets, leading to a favorable trade-off between the efficacy and computation burden. The resulting algorithms are theoretically motivated by our analysis framework and explicitly balance the plasticity and stability, mitigating catastrophic forgetting. Finally, we concentrate on experiments on the tabular setting to demonstrate our theoretical results, suggesting the applicability of our proposed framework and the potential of our proposed algorithms in real applications. Our contributions are summarized as follows:

- We propose a general analysis framework for continual RL, showing that the catastrophic forgetting is equivalent to a weighted MDPs difference regarding the optimal Q functions.

- We examine the convergence behaviors of typical continual RL baselines, including the Finetune and sequentially multi-task learning algorithms, deepening our understanding of similar continual RL algorithms.

- A theoretically motivated continual RL algorithm is thus proposed via reweighting the previous and current Bellman targets, which explicitly balances the plasticity and stability.

## 2  RELATED WORK

**Continual RL.** Continual learning (CL) (Thrun, 1995; Chen & Liu, 2018) has been one of the most important milestones on the path to building artificial general intelligence. Existing methods can be mainly classified into three groups, including rehearsal methods (Lopez-Paz & Ranzato, 2017; Chaudhry et al., 2019), regularization-based methods (Ng et al., 1999; Kirkpatrick et al., 2017; Aljundi et al., 2018) as well as parameter isolation approaches (Xu & Zhu, 2018; Mallya & Lazebnik, 2018). However, it is less studied about how to develop suitable CL methods into RL setting (Khetarpal et al., 2022). Even only a few benchmarks have been recently proposed (Wolczyk et al., 2021; Henderson et al., 2017; Platanios et al., 2020), which still need to be verified widely. Existing continual RL algorithms are designed from a variety of perspectives, including the synaptic model (Kaplanis et al., 2018), behavioral cloning that queries all previous policies (Wolczyk et al., 2022), sparse prompting (Yang et al., 2023) and policy subspace bulding (Gaya et al., 2023). In summary, the design of continual RL algorithms seeks a trade-off between the performance and model size (Gaya et al., 2023). However, most of these continual RL approaches tend to be heuristic, and there still lacks a theoretical analysis framework in a fundamental way.

**Ensemble and Reweighted Methods in RL.** Reweighted and ensemble methods have suggested huge success in a wide range of RL problems, including the weighted Q learning (Cini et al., 2020) that can reduce the bias in target estimates, Anderson Acceleration (Walker & Ni, 2011) that reweights previous target estimates in the fixed-point iteration to speed up the convergence (Sun et al., 2021; Li, 2021) and ensemble RL (Lee et al., 2021) that helps reduce the variance in target estimates. Reweighting past data was used to search for a policy that maximizes future performance in one single non-stationary MDP (Chandak et al., 2020), while our work is the first to explore the efficacy of reweighting target estimates in continual RL setting to the best of our knowledge.

## 3  ANALYSIS FRAMEWORK FOR CONTINUAL RL

### 3.1  CONTINUAL RL SETTING AND MDPS DIFFERENCE

Consider we have s sequence of $T$ tasks denoted $t = 1, ..., T$, where each task $t$ is modeled by a Markov Decision Process (MDP) $\mathcal{M}_i = \langle \mathcal{S}_i, \mathcal{A}_i, \mathcal{P}_i, R_i, \gamma \rangle$, with a set of states $\mathcal{S}_i$ and actions $\mathcal{A}_i$, the environment transition dynamics $P_i : \mathcal{S}_i \times \mathcal{A}_i \to \mathcal{P}(\mathcal{S}_i)$ and the reward function $R_i : \mathcal{S}_i \times \mathcal{A}_i \to \mathbb{R}$. In our work, we assume the same state and action space across $T$ tasks. In the classical continual RL setting, we are seeking a global policy after a sequential training manner that can generalize favorably across all tasks. When training on each task $t$, we normally require a

*training budget*, including a moderate model size and an allowable computation cost. For practical continual RL algorithms, having full access to interact with prior MDPs or an arbitrarily large model size is typically infeasible.

**MDPs Difference.** The MDPs difference is the foundation basis to analyze continual RL, and its desirable definition should consider the variation of both reward functions and environment transition dynamics between two MDPs. Therefore, we use the difference of their optimal Q functions $Q^*(s, a) = \max_\pi Q_\pi(s, a) = \max_\pi \mathbb{E}_\pi \left[ \sum_{k=0}^\infty \gamma^k R_{t+k+1} \mid S_t = s, A_t = a \right]$ in the definition.

**Definition 1.** *(MDPs Difference) For two MDPs $\mathcal{M}_1$, $\mathcal{M}_2$ with $(\mathcal{S}, \mathcal{A}, R_1, P_1, \gamma)$, $(\mathcal{S}, \mathcal{A}, R_2, P_2, \gamma)$ and the optimal Q function $Q_1^*, Q_2^*$, the p-norm MDPs difference $d_p(M_1, M_2)$ is defined as*

$$d_p(M_1, M_2) = \|Q_1^* - Q_2^*\|_p = \left( \sum_{s,a} |Q_1^*(s, a) - Q_2^*(s, a)|^p \right)^{1/p}. \tag{1}$$

Unless otherwise stated, we mainly consider $d_2$ in our analysis. We further define a weighted version of $d_2$, denoted as $\bar{d}_p$, which has an underlying connection with catastrophic forgetting in RL later.

**Definition 2.** *(Weighted MDPs Difference) The squared 2-norm weighted MDPs difference $\bar{d}_2$ is defined as $\bar{d}_2^2(M_1, M_2) = \|Q_1^* - Q_2^*\|_w^2 = \sum_{s,a} w(s, a) \left( Q_1^*(s, a) - Q_2^*(s, a) \right)^2$ with a weight function $w(s, a)$ for each $s, a$, where $\sum_{s,a} w(s, a) = 1$.*

### 3.2 Catastrophic Forgetting in Continual RL

To have a rigorous and reasonable definition of catastrophic forgetting, we extend the widely accepted definition in deep learning scenario (Doan et al., 2021) to a value-based RL setting. For completeness, we have a brief recap about the definition of *distribution drift* and *catastrophic forgetting* in Appendix A. Assume one MDP as a probabilistic model, we have the following definition:

**Definition 3.** *(Drift between two MDPs) We denote $\widehat{Q}_S$ and $\widehat{Q}_T$ as estimated Q function after training from the source MDP $\mathcal{M}_S$ and the target MDP $\mathcal{M}_T$ by any algorithm. The resulting target policy $\pi_T$ is thus obtained following the greedy rule, i.e., $\pi_T(\cdot|s) = \arg\max_a \widehat{Q}_T(s, a)$, Thus, the drift $\delta_{S,T}^{\pi_T}(\mathcal{M}_S)$ with respect to $\pi_T$ between two MDPs is defined as:*

$$\delta_{S,T}^{\pi_T}(\mathcal{M}_S) = \left( \sum_a \pi_T(a|s) \left( \widehat{Q}_S(s, a) - \widehat{Q}_T(s, a) \right)^2 \right)_{s \in |S|}, \tag{2}$$

where the drift term $\delta_{S,T}^{\pi_T}(\mathcal{M}_S)$ can be interpreted as the Q function difference weighted by the target policy $\pi_T$ that determines the proportion of the action $a$. Note that if we apply a policy $\pi$ to explore a specified MDP, the generated trajectories $\{s, a, r, s'\}$ and the resulting state distribution $\mu^\pi$ are determined by the policy $\pi$ and the corresponding environment dynamics $P$. Based on this fact, we have the following definition of catastrophic forgetting in RL.

**Definition 4.** *(Catastrophic Forgetting between two MDPs) We denote the state distribution $\mu_S^{\pi_T}$ obtained by applying the target policy $\pi_T$ into the source MDP $\mathcal{M}_S$. The catastrophic forgetting $\Delta_{S,T}^{\pi_T}(\mathcal{M}_S)$ in RL under $\ell_2$ norm is defined as*

$$\Delta_{S,T}^{\pi_T}(\mathcal{M}_S) = \|\delta_{S,T}^{\pi_T}(\mathcal{M}_S)\|_{\mu_S^{\pi_T}} = \sum_s \sum_a \mu_S^{\pi_T}(s)\pi_T(a|s) \left( \widehat{Q}_S(s, a) - \widehat{Q}_T(s, a) \right)^2. \tag{3}$$

As suggested in Eq. 3, the catastrophic forgetting in RL $\Delta_{S,T}^{\pi_T}(M_S)$ is a weighted MDP difference defined in Definition 2, with the weights $\mu_S^{\pi_T}(s)\pi_T(a|s)$ for each $s, a$ simultaneously depending on the state distribution in the source MDP $\mathcal{M}_s$ and the target policy $\pi_T$ obtained eventually.

**Catastrophic Forgetting in Continual RL.** We next extend the definition of catastrophic forgetting between two MDPs to the continual RL setting across a set of MDPs in a sequential training manner by only maintaining one single Q function estimator $\widehat{Q}_{T_\pi}$. Importantly, we allow the queries of Q functions $\{\widehat{Q}_t\}_{t<T}$ to construct the final Q estimator $\widehat{Q}_{T_\pi}$, and hence the final global policy $\pi$ can either result from $\widehat{Q}_T$ immediately obtained after training the $T$-th MDP ($T_\pi = T$), or $\widehat{Q}_{T+1}$ ($T_\pi = T + 1$) that additionally makes use of prior Q functions in an extra time step.

**Definition 5.** *(Catastrophic Forgetting in Continual RL) We denote our final global policy as $\pi$ constructed in the time of $T_\pi$ after sequentially training any algorithm on a series of MDP $\{\mathcal{M}_i\}_{i=1,...,T}$, where $T_\pi \in \{T, T+1\}$. The catastrophic forgetting $CF(Q_{T_\pi})$ is defined as*

$$CF(Q_{T_\pi}) = \sum_{t=1}^{T} \Delta_{t,T_\pi}^{\pi}(\mathcal{M}_t) = \sum_{t=1}^{T} \sum_{s} \mu_t^{\pi}(s) \sum_{a} \pi(a|s) \left( \widehat{Q}_t(s,a) - Q_{T_\pi}(s,a) \right)^2 . \quad (4)$$

Eq. 4 allows an optimal Q estimator $\widehat{Q}_{T_\pi}$ by minimizing $\widehat{Q}_{T_\pi} = \arg\min CF(Q_{T_\pi})$. Again, we can directly use $\widehat{Q}_T$ immediately obtained after training the $T$-th MDP ($T_\pi = T$) , in which case $CF(Q_T) = \sum_{t=1}^{T-1} \Delta_{t,T}^{\pi}(\mathcal{M}_t)$ as $\Delta_{T,T}^{\pi}(\mathcal{M}_T) = 0$. Alternatively, we can apply an extra transformation on $\{\widehat{Q}_t\}_{t=1}^{T}$ in an extra time step instead of $\widehat{Q}_T$ to construct $\widehat{Q}_{T_\pi}$ ($T_\pi = T + 1$). Finally, the global policy $\pi_{\text{CL}}$ is attained by $\pi_{\text{CL}}(\cdot|s) = \arg\max_a \widehat{Q}_{T_\pi}(s,a)$ based on the greedy rule.

## 4 BASELINE ANALYSIS: LOWER AND UPPER BOUND ALGORITHMS

Based on the framework in Section 3, we next examine behaviors of two typical continual RL baselines, which are regarded as one lower and upper bound that continual RL algorithms can achieve.

### 4.1 FINETUNE ALGORITHM: A LOWER-BOUND BASELINE ($T_\pi = T$)

The Finetune algorithm follows MDP-wise training via a specific RL algorithm without imposing any strategy, indicating that the obtained Q function after training the previous MDP serves as the initialization of Q functions on the current MDP. As such, the global policy is directly developed based on $\widehat{Q}_T$ after training on the $T$-th MDP immediately. Our results show that **1) Convergence:** *the estimator $\widehat{Q}_t$ we obtain in each MDP will converge to the MDP-dependent optimal Q function $Q_t^*$ regardless of MDPs differences*, **2) Convergence rate:** *the convergence rate on the current MDP is determined by its MDP difference only with its preceding MDP.*

We present our analytical results in the general framework of Neural Fitted Q Iteration (Neural FQI) (Riedmiller, 2005; Fan et al., 2020) that provides a statistical interpretation of DQN (Mnih et al., 2015) while capturing its two key features, i.e., the leverage of target network and the experience replay: $\widehat{Q}_\theta^{k+1} = \arg\min_{Q_\theta} \frac{1}{n} \sum_{i=1}^{n} [y_i - Q_\theta(s_i, a_i)]^2$, where the target $y_i = r(s_i, a_i) + \gamma \max_{a \in \mathcal{A}} Q_{\theta^*}^k(s_i', a)$ is fixed within every $T_{\text{target}}$ steps to update target network $Q_{\theta^*}$ by letting $\theta^* = \theta$. We apply the Finetune algorithm in the continual RL setting and within each MDP, we deploy the Neural FQI. The resulting *Finetune Neural FQI* has the following convergence results:

**Theorem 1.** *(Convergence of Finetune Neural FQI.) Denote $Q_t^*$ as the optimal Q function for the $t$-th MDP after the separate training, $\widehat{Q}_t^k$ as the Q function estimate after the $k$-th phase of Neural FQI in the $t$-th MDP. If $\widehat{Q}_t^k$ is a consistent estimator, then we have:*

*(1) $\sup_{s,a} |\widehat{Q}_t^k(s,a) - Q_t^*(s,a)| \leq \gamma^k d_\infty(\mathcal{M}_{t-1}, \mathcal{M}_t)$ and $\|\widehat{Q}_t^k - Q_t^*\|_\infty \to 0$ as $k \to +\infty$.*

*(2) The iteration complexity is $\mathcal{O}(\log \frac{d_\infty(\mathcal{M}_{t-1}, \mathcal{M}_t)}{\epsilon})$ in the $t$-th MDP given the tolerance error $\epsilon$.*

See Appendix B for the detailed proof. Theorem 1 indicates the Q function estimate has a *Markov-like property*, whose convergence rate is determined by the MDP difference only between the preceding and current MDPs regardless of other MDPs. More importantly, the Q function estimator in the Finetune algorithm in each MDP would asymptotically converge to the MDP-dependent optimal Q function $Q_t^*$, suffering from the complete catastrophic forgetting about the knowledge of previous MDPs. Results in Theorem 1 about the commonly used Finetune algorithm may not be surprising, but we are the first to rigorously illuminate its convergence behavior.

### 4.2 SEQUENTIAL MULTI-TASK LEARNING: AN UPPER-BOUND BASELINE ($T_\pi = T + 1$)

The Finetune algorithm analyzed in Theorem 1 serves as the lower-bound baseline as it suffers from complete catastrophic forgetting. Next, we consider a soft upper-bound baseline called *Sequential Multi-task learning algorithm* that has access to the interaction with previous MDPs as well as their

optimal Q functions in a sequential training manner. Analyzing the behaviors of sequential multi-task learning algorithms also helps us to answer a fundamental question:

***Question****: Even if we have access to optimal Q functions for all MDPs, it is tractable to derive the optimal Q function in continual RL?*

Answering this question requires us to investigate the optimality by minimizing the catastrophic forgetting quantity $CF(Q_{T_\pi})$ in Eq. 4 when $T_\pi = T + 1$. In this circumstance, we apply the Finetune algorithm across all MDPs that helps us to access all optimal Q functions, but we also conduct a function based on all optimal Q functions, i.e., $Q^{\text{opt}} = f(\{Q_t^*\}_{t=1}^T)$, to derive an optimal Q function estimator. Interestingly, this leads to an implicit optimality equation in terms of the optimal Q function $\widetilde{Q}$ and its policy $\pi$ as shown in Proposition 1. See Appendix C for the proof.

**Proposition 1.** *(Optimality Equation for Sequential Multi-task Learning) Assume the conditions in Theorem 1 hold, the optimal estimator $\widetilde{Q}$ by minimizing $CF(Q_{T_\pi})$ in Eq. 4 satisfies the equation:*

$$\widetilde{Q}(s,a) = \sum_{t=1}^T w_t^\pi(s,a)Q_t^*(s,a) / \sum_{t=1}^T w_t^\pi(s,a), \qquad (5)$$

*where $w_t^\pi(s,a) = \mu_t^\pi(s)\pi(a|s)$ is the weight. Following the greedy rule, $\pi(a^*|s) = 1$ if $a^* = \arg\max_{a'} \widetilde{Q}(s,a')$, otherwise, $\pi(\cdot|s) = 0$.*

**Approximation of $w_t^\pi(s,a)$.** For practical algorithms, the weight $w_t^\pi(s,a)$ can be approximated by Monte Carlo, i.e., $\widehat{w}_t^\pi(s_i, a_i) = \frac{1}{N_t}\sum_{i=1}^{N_t} \mathbf{1}_{\{s_t=s_i, a_t=a_i\}}$, where $(s_i, a_i) \sim \rho_t^\pi$ is drawn by applying the policy $\pi$ in the t-th MDP, and $\rho_t^\pi$ is the resulting steady state-action distribution. $N_t$ is the number of Monte Carlo simulations and $s_t, a_t$ are in the t-th MDP. Thus, the approximated optimal continual Q estimator $\widehat{\widetilde{Q}}$ can be expressed as $\widehat{\widetilde{Q}}(s_i, a_i) = \sum_{t=1}^T \widehat{w}_t^\pi(s_i, a_i)Q_t^*(s_i, a_i) / \sum_{t=1}^T \widehat{w}_t^\pi(s_i, a_i)$.

**Online Alternating algorithm.** Although the optimal continual Q estimator holds a weighted mean form in terms of $\{Q_t^*\}$, where $t = 1, ..., T$, it is an *implicit equation* regarding $\widetilde{Q}(s,a)$ as $w_t^\pi$ in the RHS of Eq. 5 is also determined by $\widetilde{Q}(s,a)$, i.e., a coupling relationship between $\pi$ and $Q$. To solve this equation, we introduce an online alternating algorithm (Algorithm 1) that alternately updates the optimal Q function and its resulting policy. However, the algorithm requires the interaction with all $T$ MDPs to have an accurate evaluation of the weight $w_t^\pi(s,a)$ for each MDP across $t = 1, .., T$, which is typically intractable in computation, especially for MDPs with large state and action spaces. We defer Algorithm 1 with the detailed description to Appendix D.

## 5    CONTINUAL RL VIA REWEIGHTING BELLMAN TARGETS

**Motivation.** Based on the baseline analysis in Section 4, the lower-bound baseline, the finetune algorithm, will lead to catastrophic forgetting, while the upper-bound baseline, the sequential multi-task learning, can result in an optimal Q function estimator in a weighted form of all optimal Q function, at the cost of an almost intractable computation burden. This motivates us to find a trade-off between the two baselines to balance the computation cost and catastrophic forgetting. *A key insight is the optimal Q function $\widetilde{Q}$ by minimizing the catastrophic forgetting turns out to be a weighted average of the optimal Q functions for all MDPs as suggested in Eq. 5*, it is therefore theoretically principled to design a continual RL algorithm by reweighting Bellman targets, e.g., the (optimal) Q functions for MDPs that the agent has interacted. In addition, our algorithm also explicitly considers the crucial two characteristics of continual learning, i.e., stability and plasticity:

**Stability.** Incorporating past Bellman targets, e.g., the target Q functions, in a reweighted form can guide the current Q function to reuse the knowledge of previous MDPs, maintaining stability.

**Plasticity.** Incorporating the current Bellman target in the current MDP is necessary to guarantee the plasticity and helps the algorithm to converge.

### 5.1    ALGORITHM FRAMEWORK BY REWEIGHTING BELLMAN TARGETS

Recap the updated rule $Q_{k+1}(s,a) \leftarrow Q_k(s,a) + \eta_k[r(s,a) + \gamma\max_{a'}Q_k(s',a') - Q_k(s,a)]$ in the vanilla Q learning, where $\eta_k$ is the step size in the k-th step. Similarly, we express the

updating rule in continual Q learning for the t-th MDP via reweighted targets as $Q_k^t(s,a) \leftarrow Q_k^t(s,a) + \eta_k \left[ r^t(s,a) + \gamma \sum_{i=1}^t \alpha_i \max_{a'} \widehat{Q}_i(s',a') - Q_t^t(s,a) \right]$, where we use the weighted target $\sum_{i=1}^t \alpha_i \max_{a'} \widehat{Q}_i(s',a')$ with the weight $\alpha_i$ plus the reward $r^t(s,a)$ collected in the current $t$-th MDP as the target. In the Neural FQI framework, we further have

$$\widehat{Q}_{\theta_t}^{k+1} = \underset{Q_{\theta_t}^k}{\operatorname{argmin}} \frac{1}{n} \sum_{i=1}^n \left[ \bar{y}_i^t(\alpha) - Q_{\theta_t}^k(s_i, a_i) \right]^2, \tag{6}$$

where the reweighted target is $\bar{y}_i^t(\alpha) = r^t(s_i, a_i) + \gamma \sum_{j=1}^t \alpha_j \max_{a'} \widehat{Q}_j^k(s_i', a')$. $\widehat{Q}_t^k = Q_{\theta_t^*}^k$ is the target network in the t-th MDP, serving as the target for the current MDP. The resulting algorithm in Eq. 6 is a variant of Neural FQI as the reweighted target is fixed within each phase of updating.

**Principle of Selecting Optimal $\alpha$ by Balancing Stability and Plasticity.** Directly conducting the algorithm in Eq. 6 may neither lead to the convergence on the t-th MDP, nor minimize the catastrophic forgetting. We thus choose the optimal $\alpha$ by considering: **(1) Stability:** to minimize catastrophic forgetting $\text{CF}(Q_{T_\pi})$ while incorporating the knowledge from the previous MDPs, and **(2) Plasticity:** the convergence guarantee in the current MDP. This involves a fundamental trade-off as relying on previous knowledge (stability) with an overly small $\alpha_t$ tends to yield the divergence issue (plasticity) on the current $t$-th MDP of the continual RL algorithm.

## 5.2 STABILITY: MINIMIZING THE CATASTROPHIC FORGETTING

With a pre-specified weighted form of $\widehat{Q}_{T_\pi}$ in Eq. 4, the optimized $\alpha^*$ that minimizes $\text{CF}(Q_{T_\pi})$ will leads to $\bar{y}_i^t(\alpha^*)$ and a *bi-level optimization* in the Neural FQI framework:

$$\widehat{Q}_{\theta_t}^{k+1}(\alpha^*) = \underset{Q_{\theta_t}^k}{\operatorname{argmin}} \frac{1}{n} \sum_{i=1}^n \left[ \bar{y}_i^t(\alpha^*) - Q_{\theta_t}^k(s_i, a_i) \right]^2, \text{ s.t. } \alpha^*(\widehat{Q}_t^k) = \arg\min_\alpha \sum_{i=1}^t \Delta_{i,t}^\pi(\mathcal{M}_i; \alpha),$$

where $\sum_{i=1}^t \Delta_{i,t}^\pi(\mathcal{M}_i; \alpha) = \sum_{i=1}^t \sum_s \mu_i^\pi(s) \sum_a \pi(a|s) \left( \widehat{Q}_i(s,a) - \sum_{j=1}^t \alpha_j \widehat{Q}_j^k(s,a) \right)^2$ referred to Eq. 4 by additionally replacing $\widehat{Q}_{T_\pi}$ with a pre-specified weighted form $\sum_{j=1}^t \alpha_j \widehat{Q}_j^k$. Similarly, we also denote $\widehat{Q}_t^k = Q_{\theta^*}^k$ for brevity, which is used to maintain the plasticity as analyzed later. However, the resulting lower-level optimization is also computationally expensive as the exact evaluation of $w_t^\pi(s,a) = \mu_i^\pi(s)\pi(a|s)$ in $\Delta_{i,t}^\pi(\mathcal{M}_i; \alpha)$ still requires to query and interact with the previous MDPs. As such, we choose to simply $\Delta_{i,t}^\pi(M_i; \alpha)$ by approximating $w_t^\pi(s,a)$ by a uniform distribution, i.e., $w_t^\pi(s,a) \approx \frac{1}{|S||A|}$:

$$\sum_{i=1}^t \Delta_{i,t}^\pi(\mathcal{M}_i; \alpha) \approx \sum_{i=1}^t \|\delta_i^k \alpha\|_2^2 = \|\Gamma^k \alpha\|_2^2, \tag{7}$$

where $\delta_i^k = [\widehat{Q}_i^k - \widehat{Q}_1^k, \widehat{Q}_i^k - \widehat{Q}_2^k, ..., \widehat{Q}_i^k - \widehat{Q}_t^k] \in \mathbb{R}^{|S \times A| \times t}$ and $\Gamma^k = [\delta_1^T, \delta_2^T, ..., \delta_t^T]^T \in \mathbb{R}^{t|S \times A| \times t}$ are the constant matrix. As $\alpha$ is also a convex combination, the lower-level optimization with a quadratic form is convex, which has a unique solution and can be effectively solved by the commonly-used optimization tool, for example, the CVXOPT toolbox in Python.

**Tractable Computation for $\Gamma^K$.** Within each batch update for practical algorithms, $|\mathcal{S}|$ in the dimension of $\Gamma^K$ is reduced to the batch size regardless of large or continuous state space. Moreover, the Q function is typically not large to avoid the instability issue (Bjorck et al., 2021), and therefore the corresponding stochastic continual RL algorithm is tractable and effective in computation.

**Comparison with Anderson Acceleration.** The optimization form in Eq. 7 is similar to the widely used *Anderson Acceleration* (Walker & Ni, 2011) technique that can speed up RL algorithms (Sun et al., 2021; Li, 2021). In Anderson Acceleration, within the Neural FQI framework, weights can be solved by $\alpha^*(\widehat{Q}_t^k) = \underset{\alpha}{\operatorname{argmin}} \left\| \sum_{i=1}^t \alpha_i \left( \mathcal{T} \widehat{Q}_t^{k+1-i}(s,a) - \widehat{Q}_t^{k+1-i}(s,a) \right) \right\|_2$, aiming at accelerating the training on the current MDP by leveraging the information from previous Q functions *in the whole iteration still in the current MDP* ranging from $Q_t^k$ to $Q_t^{k-t}$. By contrast, our method additionally uses the previous Q functions *in the previous tasks from different MDPs*.

### 5.3 PLASTICITY: CONVERGENCE GUARANTEE

We analyze the convergence of our algorithm within the dynamic programming framework. We start by introducing a *continual learning Bellman Optimality Operator* $\mathcal{T}_{\text{CL}}^{\text{opt}}$ as follows:

$$
\begin{aligned}
\mathcal{T}_{\text{CL}}^{\text{opt}} Q_t(s,a) &= \mathbb{E}\left[R(s,a)\right] + \sum_{i=1}^t \gamma \sum_{s'} \mathcal{P}_{s,s'}^a \alpha_i \max_{a'} Q_i(s',a') \\
&= \mathbb{E}\left[R(s,a)\right] + \sum_{i=1}^t \gamma \sum_{s'} \mathcal{P}_{s,s'}^{\pi^*} \alpha_i Q_i(s',a'),
\end{aligned}
\tag{8}
$$

where we let $\sum_{s'} \mathcal{P}_{s,s'}^a \sum_{a'} \pi^*(a'|s') = \sum_{s'} \mathcal{P}_{s,s'}^{\pi^*}$, omitting $a$ in $\mathcal{P}_{s,s'}^{\pi^*}$ for brevity, and $\pi^*(\cdot|s') = \arg\max Q_i(s',\cdot)$. In the Neural FQI, we have $Q_t^k(s,a) = \mathcal{T}_{\text{CL}}^{\text{opt}} Q_t^{k-1}(s,a) = \mathbb{E}\left[R(s,a)\right] + \sum_{i=1}^t \gamma \sum_{s'} \mathcal{P}_{s,s'}^{\pi^*} \alpha_i Q_i^{k-1}(s',a')$. In order to guarantee the convergence under continual learning Bellman Optimality Operator $\mathcal{T}_{\text{CL}}^{\text{opt}}$, we have the following Proposition 2 with proof in Appendix E..

**Proposition 2.** *$\mathcal{T}_{CL}^{opt}$ has a $\gamma$-linear convergence rate, $\|\mathcal{T}_{CL}^{opt} Q_t^k - Q_t^k\| \leq \gamma \|\mathcal{T}_{CL}^{opt} Q_t^{k-1} - Q_t^{k-1}\|$, if*

$$
\alpha^k = \arg\min_\alpha \|\mathcal{T}_{CL}^{opt} Q^{k-1}\alpha - Q^{k-1}\alpha^{k-1}\|,
\tag{9}
$$

*where $Q^{k-1} = [Q_1^{k-1}, ..., Q_t^{k-1}] \in \mathbb{R}^{|\mathcal{S} \times \mathcal{A}|t}$ and $\mathcal{T}_{CL}^{opt} Q^{k-1} = [\mathcal{T}_{CL}^{opt} Q_1^{k-1}, ..., \mathcal{T}_{CL}^{opt} Q_t^{k-1}]$.*

### 5.4 PUTTING ALL TOGETHER: CONTINUAL RL ALGORITHM WITH REWEIGHTED TARGETS

Combining two constraints on $\alpha_k$ in terms of both stability and plasticity, we introduce the regularization coefficient $\lambda$ to have a trade-off. The resulting continual RL algorithm with reweighted Bellman targets within the framework of Neural FQI can be expressed as follows:

$$
\begin{aligned}
Q^{k+1}(\alpha^k) &= \underset{Q_{\theta_t}}{\arg\min} \frac{1}{n} \sum_{i=1}^n \left[\bar{y}_i^t(\alpha^k) - Q_{\theta_t}^k(s_i,a_i)\right]^2, \\
\text{s.t.} \quad \alpha^k &= \arg\min_\alpha \|\mathcal{T}_{\text{CL}}^{\text{opt}} Q^k \alpha - Q^k \alpha^{k-1}\| + \lambda\|\Gamma^k \alpha\|^2, \alpha \succeq 0, \alpha^\top \mathbf{1} = 1.
\end{aligned}
\tag{10}
$$

where $\lambda$ controls the strength of catastrophic forgetting (stability) over the convergence (plasticity), which is a fundamental trade-off in continual RL. It turns out that the upper-level optimization is an iterative regression problem in terms of $Q_{\theta_t}$ given the optimal $\alpha^k$, while the lower level is a recursive ridge linear regression under the convex combination constraint in terms of $\alpha^k$. The lower-level optimization is also convex and can be solved efficiently. We can initialize with $\alpha^0 = [1/t, .., 1/t]^\top$.

**Relationship with Two Baselines.** Our proposed algorithm is motivated by the optimality equation in Proposition 1 from the upper bound baseline, sequential multi-task learning, as the optimal Q function in continual RL should be a weighted form of all optimal Q functions in each MDP. Meanwhile, without incorporating the reweighting mechanism about Bellman targets, our algorithm will degenerate to the lower-bound baseline, the Finetune algorithm.

## 6 EXPERIMENTS

We conduct our experiments to verify: (1) whether the Finetune algorithm, the lower-bound baseline, converges to the MDP-dependent optimal Q function and whether the convergence rate is negatively correlated to the MDP difference as analyzed in Theorem 1, (2) when the sequential multi-task learning (Sequential ML), the soft upper-bound baseline, can outperform the lower bound baseline, and (3) whether and when our proposed continual RL algorithm with reweighted Bellman targets performs better than other baselines. We sequentially apply different continual RL algorithms on a series of MDPs with different reward functions and environment dynamics.

**Environments.** Since our analysis and the proposed algorithm are mainly on the tabular setting, we choose to demonstrate our results on a simple MDP and the GridWorld environment. Due to the benchmark in continual RL is still less studied, we have not found any other widely accepted benchmark to test value-based continual RL algorithms. Several benchmarks, including Continual

World (Wolczyk et al., 2021) are designed for policy gradient-based RL algorithms with Soft Actor Critic (SAC) (Haarnoja et al., 2018) as the basic algorithm, which is not (directly) applicable to our theoretical results and the proposed value-based RL algorithm.

## 6.1 CONTINUAL Q LEARNING ON SIMPLE MDPS

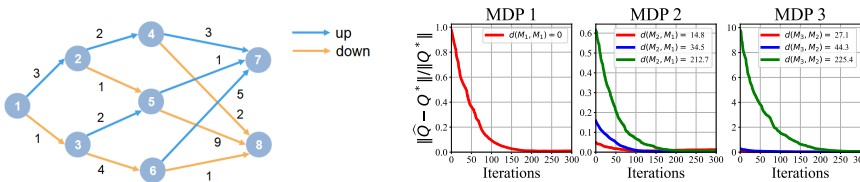

Figure 1: (**Left**) The simple MDP with different reward functions and environment dynamics. (**Right**) Learning curves of vanilla continual Q learning on three MDPs, where $Q^*$ is evaluated as the single Q learning on each MDP.

**Experimental Setup.** For the separate training on the three MDPs, where the MDP structure is in Figure 1 (Left), we denote its results as $V^*$ or $Q^*$ as the first baseline. Since this MDP can be well solved by typical searching-based algorithms, we thus evaluate the optimal value function by leveraging the Floyd algorithm denoted as "Optimal". We randomly select the reward in each edge, ranging from (0, 20), (10, 20), and (0, 30) for these three MDPs. In this MDP, when the agent takes the action "Up", it will get into the upper state with a certain probability (typically close to 1) and the other state in the reversed direction otherwise. We select these three transition probabilities as 0.9, 0.8, and 0.9, respectively. In order to demonstrate Theorem 1, we specify different MDP2 with different reward ranges, i.e., (10, 20), (30, 45), (80, 100), and plot their learning curves on Figure 1 (Right).

**Results.** As suggested in Figure 1 (Right), the learning speed increases in MDPs 2 and 3 with lines in green, blue to red lines, as we decrease incremental MDP difference, i.e., $d_\infty(\mathcal{M}_t, \mathcal{M}_{t-1})$. However, under sufficient training, the normalized Q function difference $\|\widehat{Q} - Q^*\|/\|Q^*\|$ tends to 0. These empirical results corroborate results analyzed in Theorem 1.

| Average Return | MDP1 | MDP2 | MDP3 | Average |
|---|---|---|---|---|
| Optimal | 49.0 ($\pm$3.4) | 66.3 ($\pm$2.6) | 128.0 ($\pm$33.2) | 81.1 ($\pm$11.5) |
| $V^*$ | 48.6 ($\pm$3.7) | 66.3 ($\pm$2.6) | 127.6 ($\pm$33.5) | 80.9 ($\pm$11.7) |
| Finetune | 38.3 ($\pm$5.0) | 54.9 ($\pm$5.3) | 127.8 ($\pm$33.3) | 73.7 ($\pm$11.1) |
| Sequential ML | 45.6 ($\pm$5.2) | 63.8 ($\pm$4.2) | 119.8 ($\pm$33.5) | **76.4** ($\pm$11.0) |
| Ours | 42.4 ($\pm$3.9) | 55.2 ($\pm$5.9) | 125.6 ($\pm$33.2) | 74.4 ($\pm$11.2) |

Table 1: Achieved average return of different algorithms in continual RL setting over 20 runs. The optimal policy of our method is robust to $\lambda = 0.0, 5.0, 10.0$ in this simple MDP.

In order to demonstrate the superiority of Sequential ML over the Finetune algorithm, we specify the reward in the MDP 3 as $C - \frac{1}{2}R_1 - \frac{1}{2}R_2$, where $R_1$ and $R_2$ are the reward vectors generated in the first and second MDPs and $C$ is a constant generated in the rage of (40, 80). We do 20 runs across all algorithms. Results in Table 1 show that our continual RL algorithm performs favorably in-between the Finetune and sequential multi-task learning algorithms. We also have more experiments in this setup in Appendix F

## 6.2 CONTINUAL Q LEARNING ON THE GRID WORLD

**Experimental Setup.** The GridWorld has been used to evaluate continual RL algorithms in (Kaplanis et al., 2018). The GridWorld environment is a stochastic MDP, where the same action could have different outcomes and enter different next states). In particular, the environment moves the agent in the intended direction with a certain probability $P$, and with probability $1 - P$, they move

the agent in a random other direction. We set both the width and height as 10. The state is 2-dimensional and a more detailed setting description can refer to (Kaplanis et al., 2018). We choose four MDPs with different transition probabilities $P$, i.e., 0.75, 0.8, 0.85 and 0.9. As illustrated in Figure 2 (Left), the agent will receive +100 reward if it encounters the gold, and -100 reward if it encounters a bomb. In the four MDPs, we change the locations of gold and bomb from the right-upper part to the left-lower part gradually. Thus, both reward functions and the environment transitions would be different.

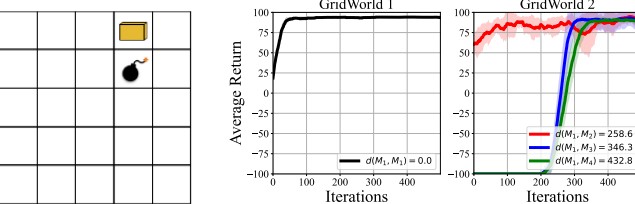

Figure 2: (**Left**) Grid World Environment. **(Right)** Convergence of the Finetune algorithm on Grid-World with the relationship of the MDP differences. We sequentially apply the Finetuen algorithm from MDP1 to the other three MDPs with different MDP differences. Results are averaged over 10 runs. We use a weighted MDP difference ($\ell_2$) by applying the obtained policy to explore the certain MDP following the $\epsilon$-greedy rule.

**Results.** We first demonstrate the conclusion in Theorem 1. As suggested in Figure 2 (Right), the learning speed decreases as we increase the MDP difference between MDP 2 and MDP1, from the red, blue to the green lines, indicating that a larger MDP difference would decrease the learning speed of Finetune algorithm. We also do 5 runs to compare the performance of different continual RL algorithms. Results in Table 2 suggest that our proposed algorithm performs favorably compared with the other baselines, in which our algorithm with $\lambda = 18.0$ performs best. Notably, the Sequential ML does not perform well as it suffers form the divergence issue, where the online alternating algorithm may not converge in the considered iteration steps.

| Average Return | MDP1 | MDP2 | MDP3 | MDP4 | Average |
|---|---|---|---|---|---|
| $V^*$ | 90.7 ($\pm$1.3) | 93.2 ($\pm$4.5) | 92.3 ($\pm$6.1) | 90.7 ($\pm$3.9) | 90.7 ($\pm$1.9) |
| Finetune | -100.0 ($\pm$0.0) | -100.0 ($\pm$0.0) | -92.3 ($\pm$8.9) | 26.2 ($\pm$87.0) | -66.5 ($\pm$23.2) |
| Sequential ML | -93.0 ($\pm$5.6) | -98.0 ($\pm$1.9) | -61.2 ($\pm$17.3) | 18.1 ($\pm$67.8) | -58.5 ($\pm$20.7) |
| Ours ($\lambda = 0.0$) | -99.0 ($\pm$0.6) | 37.4 ($\pm$53.7) | -102.8 ($\pm$3.9) | -100.0 ($\pm$0.0) | -66.1 ($\pm$12.9) |
| Ours ($\lambda = 12.0$) | -70.2 ($\pm$42.3) | 42.6 ($\pm$44.0) | -100.0 ($\pm$0.2) | -100.0 ($\pm$0.0) | -56.9 ($\pm$18.4) |
| Ours ($\lambda = 18.0$) | -48.8 ($\pm$67.3) | 84.4 ($\pm$8.6) | -98.0 ($\pm$2.2) | -99.0 ($\pm$0.2) | **-40.6** ($\pm$16.2) |

Table 2: Average return with standard deviations of different continual RL algorithms over 5 runs.

## 7 DISCUSSIONS AND CONCLUSION

In this paper, we are addressing the unsolved issues in the field of continual RL. We begin by providing a general analysis framework to characterize the catastrophic forgetting in continual RL. Based on this framework, we also conduct the baseline analysis, including the convergence results and an optimal continual Q estimator along with an online alternating algorithm. Finally, we propose a practical continual RL algorithm via reweighted targets. The resulting bi-level and ridge recursive algorithm has shown promising results in the considered experiments.

There are still some limitations in our analysis as well as our proposed approach. Firstly, we concentrate on the tabular setting with a low-dimensional state space, but we have not fully considered the impact of representation sharing for deep RL algorithms within multiple MDPs in the continual RL regime. Additionally, our continual RL setting also requires access to the task boundary and assumes the same state and action spaces across different MDPs. However, the research community could put more effort into the more challenging task-agnostic setting with arbitrarily different state and action spaces across various MDPs to get closer to real-world scenarios in the future.

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

## A  Definition of Distribution Drift and Catastrophic Forgetting

We first introduce the concept of *drift* in the process of learning a parameterized function $f$ from the source data distribution $\tau_S$ with the dataset $\mathcal{D}_{\tau_S}$ to the target data distribution $\tau_T$ with the dataset $\mathcal{D}_{\tau_T}$. After learning $f$ on the source dataset $\mathcal{D}_{\tau_S}$, we obtain the estimated function $\widehat{f}_{\tau_S}$. Then we apply the same model architecture $f$ on the target dataset $\mathcal{D}_{\tau_T}$ with any learning algorithms, and finally we evaluate the drift of the attained $\widehat{f}_{\tau_T}$ via $\delta^{\tau_S \to \tau_T}$ defined as (Doan et al., 2021):

$$\delta^{\tau_S \to \tau_T}\left(X^{\tau_S}\right) = \left(\widehat{f}_{\tau_T}(x) - \widehat{f}_{\tau_S}(x)\right)_{(x,y) \in \mathcal{D}_{\tau_S}} \tag{11}$$

Based on the definition of *drift*, we define the *vanilla catastrophic forgetting* $\Delta^{\tau_S \to \tau_T}$ as

$$\Delta^{\tau_S \to \tau_T}\left(X^{\tau_S}\right) = \|\delta^{\tau_S \to \tau_T}\left(X^{\tau_S}\right)\|_2^2 = \sum_{(x,y) \in \mathcal{D}_{\tau_S}} \left(\widehat{f}_{\tau_T}(x) - \widehat{f}_{\tau_S}(x)\right)^2, \tag{12}$$

where the catastrophic forgetting can be further simplified as $\Delta^{\tau_S \to \tau_T} = \left\|\phi\left(X^{\tau_S}\right)\left(\omega_{\tau_T}^* - \omega_{\tau_S}^*\right)\right\|_2^2$ in the Neural Tangent Kernel (NTK) regime (Doan et al., 2021; Jacot et al., 2018), allowing the proposal of new continual learning approaches. In deep learning, minimizing the catastrophic forgetting $\Delta^{\tau_S \to \tau_T}$ is equivalent to minimizing a weighted drift in terms of the prediction function $\widehat{f}$ with the weights determined by the dataset.

## B  Proof of Theorem 1

*Proof.* **Convergence.** We start from the key inequality in Neural FQI, where the Q function is parameterized via a neural network and iteratively optimized in each phase of each MDP sequentially. Since $\widehat{Q}_t^k$ is a consistent estimator, it converges to its expectation form in probability. For simplicity, we consider their expectation form in our proof. We denote $Q_t^{k+1}(s,a) = \mathbb{E}\left[r(s,a)\right] + \gamma \max_{a \in \mathcal{A}} \mathbb{E}_{s'}\left[Q_{\theta^*}^k\left(s',a\right)\right] = \mathcal{T}^{\text{opt}}Q_{\theta^*}^k(s,a)$ in the asymptotic case ($n \to +\infty$) as the target Q that guides to estimator $\widehat{Q}_t^{k+1}$. Since $Q_{\theta^*}^k = \widehat{Q}_t^k$ when periodically updating the target network, we have $Q_t^k(s,a) = \mathcal{T}^{\text{opt}}Q_{\theta^*}^{k-1}(s,a) = \mathcal{T}^{\text{opt}}\widehat{Q}_t^{k-1}(s,a)$. Thus, we have

$$\sup_{s,a}\left|\widehat{Q}_t^k(s,a) - Q_t^*(s,a)\right|$$

$$\leq \sup_{s,a}\left|\widehat{Q}_t^k(s,a) - Q_t^k(s,a)\right| + \sup_{s,a}\left|Q_t^*(s,a) - Q_t^k(s,a)\right|$$

$$= \sup_{s,a}\left|\widehat{Q}_t^k(s,a) - Q_t^k(s,a)\right| + \sup_{s,a}\left|\mathcal{T}^{\text{opt}}Q_t^*(s,a) - \mathcal{T}^{\text{opt}}\widehat{Q}_t^{k-1}(s,a)\right|$$

$$\leq \sup_{s,a}\left|\widehat{Q}_t^k(s,a) - Q_t^k(s,a)\right| + \gamma \sup_{s,a}\left|\widehat{Q}_t^{k-1}(s,a) - Q_t^*(s,a)\right|$$

$$\leq \sup_{s,a}\left|\widehat{Q}_t^k(s,a) - Q_t^k(s,a)\right| + \gamma \sup_{s,a}\left|\widehat{Q}_t^{k-1}(s,a) - Q_t^{k-1}(s,a)\right| + \gamma^2 \sup_{s,a}\left|\widehat{Q}_t^{k-2}(s,a) - Q_t^*(s,a)\right|$$

$$\cdots$$

$$\leq \sum_{i=0}^{k-1} \gamma^i \sup_{s,a}\left|\widehat{Q}_t^{k-i}(s,a) - Q_t^{k-i}(s,a)\right| + \gamma^k \sup_{s,a}\left|\widehat{Q}_t^0(s,a) - Q_t^*(s,a)\right|$$

$$\leq \sum_{i=0}^{k-1} \gamma^i \alpha + \gamma^k \sup_{s,a}\left|\widehat{Q}_{t-1}(s,a) - Q_t^*(s,a)\right|$$

$$\leq \frac{1}{1-\gamma}\alpha + \gamma^k \sup_{s,a}\left|\widehat{Q}_{t-1}(s,a) - Q_t^*(s,a)\right|$$

$$\to \gamma^k \sup_{s,a}\left|\widehat{Q}_{t-1}(s,a) - Q_t^*(s,a)\right| \quad \text{(no optimization error)}$$

$$\to 0 \quad (k \to +\infty)$$

$$\tag{13}$$

where $Q_t^1(s,a) = r(s,a) + \gamma \max_{a \in \mathcal{A}} \widehat{Q}_t^0(s',a)$ in the target estimate in the first phase for the $t$-th MDP, and $\widehat{Q}_t^0 = \widehat{Q}_{t-1}$ is the estimated Q function after $t-1$-th MDP. We also introduce an upper bound $\alpha$ of each Neural FQI iteration, which tends to 0 in an ideal case. This is because the experience buffer induces independent samples $\{(s_i, a_i, r_i, s_i')\}_{i \in [n]}$ and ideally without the optimization and TD approximation errors, the updating in each phase of Neural FQI is exactly the updating under Bellman optimality operator (Fan et al., 2020). This implies that $\|\widehat{Q}_t^k - Q_t^k\|_\infty \to 0$ ideally, and thus their upper bound $\alpha$ converges to 0. The last arrow of proof holds when $k \to +\infty$ as long as $\|\widehat{Q}_{t-1} - Q_t^*\|_\infty$ is bounded or not arbitrarily large.

Putting all together, the proof above indicates that $\|\widehat{Q}_t^k - Q_t^*\|_\infty \to 0$ as $k \to +\infty$ without the optimization error in each Neural FQI **regardless of the initialization** $\widehat{Q}_{t-1}$. As $\widehat{Q}_1^k \to Q_1^*$ under the above conditions, we can easily prove in a recursive way that $\widehat{Q}_t^k \to Q_t^*$ for each $t = 1, ..., T$ in this Finetune algorithm, i.e., $\widehat{Q}_t = Q_t^*$. As such, we further plug this condition into the last two arrows in the proof above, and therefore, given the $k$-th phase in the $t$-th MDP, we have

$$\sup_{s,a} |\widehat{Q}_t^k(s,a) - Q_t^*(s,a)| \leq \gamma^k \sup_{s,a} \left| Q_{t-1}^*(s,a) - Q_t^*(s,a) \right|$$
$$= \gamma^k d_\infty(\mathcal{M}_{t-1}, \mathcal{M}_t), \tag{14}$$

where the RHS in terms of Q function is exactly the MDPs difference we define in Definition 1.

**Iteration complexity.** Let RHS in Eq. 14 be less than $\epsilon$, after taking log transformation, we have:

$$k \log \gamma + \log d_\infty(\mathcal{M}_{t-1}, \mathcal{M}_t) \leq \log \epsilon$$

Finally, we have:

$$k \geq C \log \frac{d_\infty(\mathcal{M}_{t-1}, \mathcal{M}_t)}{\epsilon}, \tag{15}$$

where $C = -\frac{1}{\log \gamma}$. This indicates that the iteration complexity is $\mathcal{O}(\log d_\infty(\mathcal{M}_{t-1}, \mathcal{M}_t)/\epsilon)$ given an $\epsilon$ iteration error. In other words, a larger MDP difference between the current and preceding ones would require a larger number of iterations for $\widehat{Q}_t^k$ in order to converge to MDP-dependent optimal $Q_t^*$ in the t-th MDP.

$\square$

## C    PROOF OF PROPOSITION 1

*Proof.* Under the conditions of Theorem 1, where $T_\pi = T$, we have $\widehat{Q}_T = Q_T^*$, which suffers from the complete catastrophic forgetting and is typically not the optimal continual minimizer. By contrast, with full access to all MDPs, we can further construct a mapping based on previous optimal Q functions in each MDP ($T_\pi = T + 1$) for an optimal one in the sequential multi-task learning. By plugging $\widehat{Q}_t = Q_t^*$, the objective function of Eq. 4 can be simplified as follows

$$\mathrm{CF}(Q) = \sum_{t=1}^{T} \sum_s \mu_t^\pi(s) \sum_a \pi(a|s) \left( Q_t^*(s,a) - Q(s,a) \right)^2$$
$$= \sum_{t=1}^{T} \sum_{s,a} w_t^\pi(s,a) \left( Q_t^*(s,a) - Q(s,a) \right)^2 \tag{16}$$

where $w_t^\pi(s,a) = \mu_t^\pi(s)\pi(a|s)$. Since we hope to find the optimal Q estimator in terms of the whole objective function, the Q function mapping from $s, a$ to the Q is just the inner mapping, we only need to consider the optimality equation for a specific $s, a$. Although $w_t^\pi(s,a)$ is coupled with $Q(s,a)$ following the greedy rule in terms of the policy $\pi$, when we fix $w_t^\pi$ within a bi-level optimization, the objective function above is equivalent to a quadratic function regarding $Q(s,a)$. By taking the derivative of $\mathrm{CF}(Q)$ regarding $Q(s,a)$ and then let the gradient equal to 0, for each $s, a$, it arrives at

$$\sum_{t=1}^{T} w_t^\pi(s,a) \left( Q_t^*(s,a) - Q(s,a) \right) = 0 \tag{17}$$

We finally have:

$$\widetilde{Q}(s,a) = \sum_{t=1}^{T} w_t^{\pi}(s,a) Q_t^*(s,a) / \sum_{t=1}^{T} w_t^{\pi}(s,a) \quad \forall s,a, \tag{18}$$

where the restraint is $w_t^{\pi}(s,a) = \mu_t^{\pi}(s)\pi(a|s)$ with $\pi(a^*|s) = 1$ if $a^* = \arg\max_{a'} \widetilde{Q}(s,a')$, otherwise $\pi(a|s) = 0$.

$\square$

## D  ONLINE ALTERNATING ALGORITHM FOR OPTIMALITY EQUATION OF SEQUENTIAL MULTI-TASK LEARNING

---

**Algorithm 1** Online Alternating Algorithm for Sequential Multi-task Learning

---

1: Given the $\{Q_t^*\}$ for $t = 1, .., T$, and initialize $Q^{(0)}$. Set the total training steps $K$, evaluation step $L$, and the number of samples $N_t$ for each $t = 1, ..., T$. Initialize $l = 1$.
2: **while** $l \leq L$ **do**
3:    / * *Step 1: Weight Evaluation via $\pi$ determined by $Q^{(l-1)}$* * /
4:    **for** $t = 1$ to $T$ **do**
5:       Observe the initial state $s_0$ in the t-th MDP;
6:       **for** $i = 1$ to $N_t$ **do**
7:          Select $a_i = \arg\max_a Q^{(l-1)}(s_i, a)$ via the greedy rule.
8:          Perform the action $a_i$ on t-th MDP, obtain $r_i$ and $s_{i+1}$.
9:          Store the transition $(s_i, a_i)$ in the $t$-th buffer.
10:       **end for**
11:       Estimate $w_t^{\pi}$ based on samples in the $t$-th buffer:

$$\widehat{w}_t^{\pi}(s,a) \leftarrow \frac{1}{N_t} \sum_{i=1}^{N_t} \mathbf{1}_{\{s_t = s_i, a_t = a_i\}}$$

12:    **end for**
13:    / * *Step 2: Q Function Updating* * /
14:    Sample the batch of transitions $(s_i, a_i)$ from all $T$ buffers.
15:    Update Q Function for each $(s_i, a_i)$ via

$$Q^{(l)}(s_i, a_i) \leftarrow \sum_{t=1}^{T} \widehat{w}_t^{\pi}(s_i, a_i) Q_t^*(s_i, a_i) / \sum_{t=1}^{T} \widehat{w}_t^{\pi}(s_i, a_i)$$

16:    $l \leftarrow l + 1$
17: **end while**

---

The algorithm procedure includes two iterative steps. The first one is the weight evaluation while fixing the $\widetilde{Q}$, in which case, the policy is also fixed. The weight evaluation $\widehat{w}_t^{\pi}(s,a)$ is typically proceeding via Monte Carlo method, and a larger number of simulations leads to a more accurate evaluation at the cost of the more computational cost. The second step is to update the Q function with the evaluated weights $\widehat{w}_t^{\pi}(s,a)$ in step one. This procedure requires the online interaction with all MDPs, and hence it is called an online alternating algorithm. This alternating optimization in Algorithm 1 is also similar to the policy/value iteration algorithm that interacts between the Q function and the policy. In our implementation, we select a sufficiently large $L$ to guarantee a favorable convergence of our online alternating algorithm, although an overly large $L$ will increase the computational cost significantly.

In summary, in order to leverage all optimal Q functions to seek an optimal Q function estimator in continual RL, we are required to interact with all MDPs. This additional computation is likely to be useful in continual learning provided that the weights $\widehat{w}_t^{\pi}(s_i, a_i)$ can be approximated favorably via the online alternating algorithm within the computational budget.

## E  PROOF OF PROPOSITION 2

Directly considering two $Q_t(s,a)$ to derive the contraction mapping would be difficult as $\alpha$ depends on the $Q_t(s,a)$ and decoupling $\alpha$ and $Q_t(s,a)$ is hard. Instead, we consider the following

convergence rate proof by following (Li, 2021):

$$\|\mathcal{T}_{\mathrm{CL}}^{\mathrm{opt}}Q_t^k - Q_t^k\|$$

$$= \|\mathbb{E}\left[R(s,a)\right] + \sum_{i=1}^{t}\gamma\sum_{s'}\mathcal{P}_{s,s'}^{\pi^k}\alpha_i^k Q_i^k(s',a') - Q_t^k(s,a)\|$$

$$= \|\sum_{i=1}^{t}\alpha_i^k\left(\mathbb{E}\left[R(s,a)\right] + \gamma\sum_{s'}\mathcal{P}_{s,s'}^{\pi^k}Q_i^k(s',a') - Q_t^k(s,a)\right)\|$$

$$= \gamma\|\sum_{i=1}^{t}\alpha_i^k\left(\sum_{s'}\mathcal{P}_{s,s'}^{\pi^k}Q_i^k(s',a') - \sum_{j=1}^{t}\sum_{s''}\mathcal{P}_{s,s''}^{\pi^{k-1}}\alpha_j^{k-1}Q_i^{k-1}(s'',a'')\right)\|$$

$$= \gamma\|\sum_{i=1}^{t}\alpha_i^k\sum_{j=1}^{t}\alpha_j^{k-1}\left(\sum_{s'}\mathcal{P}_{s,s'}^{\pi^k}\mathcal{T}_{\mathrm{CL}}^{\mathrm{opt}}Q_i^{k-1}(s',a') - \sum_{s''}\mathcal{P}_{s,s''}^{\pi^{k-1}}Q_i^{k-1}(s'',a'')\right)\|$$

$$\leq \gamma\|\sum_{i=1}^{t}\alpha_i^k\sum_{j=1}^{t}\alpha_j^{k-1}\left(\sum_{s'}\mathcal{P}_{s,s'}^{\pi^*}\mathcal{T}_{\mathrm{CL}}^{\mathrm{opt}}Q_i^{k-1}(s',a') - \sum_{s''}\mathcal{P}_{s,s''}^{\pi^*}Q_i^{k-1}(s'',a'')\right)\| \tag{19}$$

$$= \gamma\|\sum_{i=1}^{t}\alpha_i^k\sum_{j=1}^{t}\alpha_j^{k-1}\left(\sum_{s'}\mathcal{P}_{s,s'}^{\pi^*}\mathcal{T}_{\mathrm{CL}}^{\mathrm{opt}}Q_i^{k-1}(s',a') - \sum_{s'}\mathcal{P}_{s,s'}^{\pi^*}Q_i^{k-1}(s',a')\right)\|$$

$$\leq \gamma\sum_{s}\mathcal{P}_{s,s'}^{\pi^*}\|\sum_{i=1}^{t}\alpha_i^k\sum_{j=1}^{t}\alpha_j^{k-1}\left(\mathcal{T}_{\mathrm{CL}}^{\mathrm{opt}}Q_i^{k-1}(s',a') - Q_i^{k-1}(s',a')\right)\|$$

$$\leq \gamma\|\sum_{i=1}^{t}\alpha_i^k\sum_{j=1}^{t}\alpha_j^{k-1}\left(\mathcal{T}_{\mathrm{CL}}^{\mathrm{opt}}Q_i^{k-1}(s',a') - Q_i^{k-1}(s',a')\right)\|$$

$$= \gamma\|\mathcal{T}_{\mathrm{CL}}^{\mathrm{opt}}Q^{k-1}\alpha^k - Q^{k-1}\alpha^{k-1}\|$$

$$\leq \gamma\|\mathcal{T}_{\mathrm{CL}}^{\mathrm{opt}}Q_t^{k-1} - Q_t^{k-1}\| \quad \text{\textcolor{red}{(need to be guaranteed)}}$$

where we define $\mathcal{T}_{\mathrm{CL}}^{\mathrm{opt}}Q_i^k = Q_i^k$ if $i < t$ and $\mathcal{T}_{\mathrm{CL}}^{\mathrm{opt}}Q^k$ indicates that we apply the operator $\mathcal{T}_{\mathrm{CL}}^{\mathrm{opt}}$ on the vector $Q^k = \{Q^k\}_{i=1}^{t}$ in an column-wise way. The first inequality we leverage the inequality $\max_x f(x) - \max_y g(y) = f(x^*) - g(y^*) \leq f(x^*) - g(x^*) \leq \max_x f(x) - g(x)$ if we assume the difference is positive without loss of generality, and $\sum_i b_i f_i - \sum_j b_j g_j = \sum_i b_i(f_i - g_i)$. $Q^k = [Q_1^k, ..., Q_t^k]$. Thus, in order to guarantee the $\gamma$-linear convergence rate, i.e.,

$$\|\mathcal{T}_{\mathrm{CL}}^{\mathrm{opt}}Q_t^k - Q_t^k\| \leq \gamma\|\mathcal{T}_{\mathrm{CL}}^{\mathrm{opt}}Q_i^{k-1} - Q_i^{k-1}\| \tag{20}$$

we need to construct the following constraint on $\alpha^k$ recursively as follows:

$$\alpha^k = \arg\min_{\alpha}\|\mathcal{T}_{\mathrm{CL}}^{\mathrm{opt}}Q^{k-1}\alpha - Q^{k-1}\alpha^{k-1}\|, \tag{21}$$

where we have $\|\mathcal{T}_{\mathrm{CL}}^{\mathrm{opt}}Q^{k-1}\alpha^k - Q^{k-1}\alpha^{k-1}\| \leq \|\mathcal{T}_{\mathrm{CL}}^{\mathrm{opt}}Q^{k-1}\alpha^{k-1} - Q^{k-1}\alpha^{k-1}\| \leq \|\mathcal{T}_{\mathrm{CL}}^{\mathrm{opt}}Q_t^{k-1}\alpha_t^{k-1} - Q_t^{k-1}\alpha_t^{k-1}\| = \alpha_t^{k-1}\|\mathcal{T}_{\mathrm{CL}}^{\mathrm{opt}}Q_t^{k-1} - Q_t^{k-1}\| \leq \|\mathcal{T}_{\mathrm{CL}}^{\mathrm{opt}}Q_t^{k-1} - Q_t^{k-1}\|$ and the second inequality is due to the definition $\mathcal{T}_{\mathrm{CL}}^{\mathrm{opt}}Q_i^k = Q_i^k$ if $i < t$.

## F EXPERIMENTS ON MDP

Further, we investigate when "Upper" is significantly superior to "Lower". Instead of randomly assigning rewards in a certain range in the previous setting, we construct a "reverse" reward function setting in the third MDP in contrast to the reward functions in the first and second MDPs, respectively. In particular, given the reward function in the first and second MDP, we set rewards in the third MDP as $C - r_1$, $C - r_2$ and $C - r_1 - r_2$, where $c$ is a pre-specified constant. It is expected that if the reward function in the three MDPs is $C - r_1$, for example, the "Lower" algorithm will "overfit" to the last MDP, which is reversed to the reward distribution in the first MDP. Therefore, its performance on the first MDP would be undesirable and worse than "Upper" that simultaneously considers

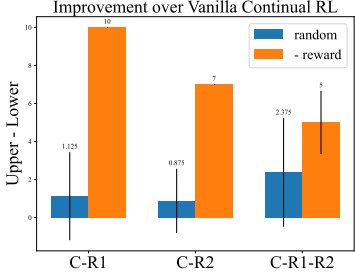

Figure 3: The first two orange bars for $C - r_1$, $C - r_2$ are calculated on MDP1 and MDP2, respectively. The third bar for $C - r_1 - r_2$ is averaged over both MDP1 and MDP2.

all MDPs. We evaluate the difference between "Upper" and "Lower" on either random (rewards are sampled randomly) and reverse reward, e.g., $C - r_1$, settings and make a detailed comparison.

As suggested in Figure 3, when the reward function in the third MDP is the reverse one of the previous MDP, "Upper" performs much better than "Lower" as "Lower" only performs based on the environment of the last MDP, which can be dramatically different from previous environments.

