# OpenReview forum: "Continual Reinforcement Learning by Reweighting Bellman Targets"
_ICLR.cc/2024/Conference — Submitted to ICLR 2024_

### Official Review · Reviewer_o16F · 2023-10-30

**Soundness:** 2 fair
**Presentation:** 3 good
**Contribution:** 1 poor
**Rating:** 3
**Confidence:** 4

**Summary:**

This paper proposed a continual reinforcement learning approach by reweighing Bellman targets under the sequential multitask setting where the algorithm has the goal of learning one Q-function for all tasks. The paper first defined a notion of catastrophic forgetting and studied the form of the optimal solution that weighted averages over the previous optimal Q functions. Following the average form, the paper proposed a weighted Bellman target algorithm and discussed how to find the weight considering two principles: stability and plasticity. The paper then conducted simulation studies on grid-world and tabular MDPs.

**Strengths:**

The proposed notion of catastrophic forgetting is novel and the analysis of the optimal form seems novel.

**Weaknesses:**

I believe the contribution of this paper to the literature is limited and many different aspects of the paper have to be significantly improved.

On a high-level, the paper assumes that the algorithm is only allowed to maintain one Q-function and it is trying to memorize all previous tasks. I think this is a too trivial setting to study. Many papers in the literature have studied more comprehensive setups where the algorithms either try to adaptively memorize tasks or maintain different Q functions for tasks that are too discrepant.

The theoretical analysis is trivial and not fruitful. The theorem 1 in section 4.1 is a direct application of the previous work and the main message is simply that fine-tuning can quickly adapt to the current task, thus, forgetting about the previous ones. Proposition 1 though seems to be interesting lacks further discussion (see my questions for detailed discussion).

The connection between the practical algorithm defined in 5.1 and the optimal solution defined in 4.2 needs to be more formally discussed. The paper considers a setting, where we want to find one global Q that works well in all MDPs. It is obvious that the optimal solution should take certain form of weighted optimal Q_t’s. What is special in equation (5) is that this weight is given by an equation with special form. I don’t see this being carried to the design of the practical algorithms. In the current form, the theory and the practical algorithm are rather irrelevant.

The writings have to be improved too. There are multiple typos and undefined notations.

**Questions:**

1. Therefore, we use the different of their optimal Q functions: You mentioned that your goal is to consider the variations in both transition and rewards. However, optimal Q function does not uniquely identify a pair of transition and rewards.

2. What do you mean by “Assume one MDP as a probabilistic model”

3. Definition 1 and 2 have overlap. You may define weighted MDPs Difference for any p norm, then Definition 1 is subsumed by Definition 2 by choosing the uniform weight

4. What is $T_{\pi}$ here? I vaguely understand what the authors try to deliver here from the word descriptions. But I believe the notations in this paragraph only makes it more confusing. How do you make queries on the previous \hat Q_{t} if this is under “a sequential training manner by only maintaining one single Q function estimator”?

5. Equ 5, I suggest an extra notation on \pi to make its dependence on \tilde{Q} explicit

6. From a theoretical point of view, the authors might want to discuss/study the cases when fine-tune can be significantly worse than Sequential multitask learning

7. It seems to me that another naive yet more meaningful baseline is to take simply take average over previous \hat Q_t

8. How do we draw Monte Carlo samples from \rho_t^{\pi} when \pi itself is unknown?

9. Do you normalize \alpha in (7)? Otherwise, this seems to lead a trivial solution

10. For tabular MDP case, the optimal algorithm (an approximate version) can obviously be implemented and compared. I do not see a reason that the authors choose not to compare.

Typos or undefined notations:
1. Consider we have s sequence of T tasks denoted t —> Consider we have a sequence of T tasks denoted by t
2. What is the index $i$ for?
3. Need to introduce \mathcal{P}(\mathcal{S}_i) notation

---

### Official Review · Reviewer_EmDp · 2023-11-01

**Soundness:** 2 fair
**Presentation:** 2 fair
**Contribution:** 3 good
**Rating:** 5
**Confidence:** 3

**Summary:**

The paper tackles the field of continual RL and presents a new tools of analysis called MDPs difference, which computes the difference between two follow-up MDPs in the continual task. This tool of reasoning is then employed to characterize the phenomenon of "catastrophic forgetting" and also provide a theoretical analysis on convergence behavior for some state-of-the-art algorithms. Furthermore, a new algorithm is presented and evaluated on two very small examples.

**Strengths:**

The paper tackles a very interesting and relevant field and manages to grasp a lot of the current issues within its scope. The approach presented in Def. 1-3 is simple and beautiful and should probably be pursued further. Providing an empirical study is commendable.

**Weaknesses:**

The paper lacks focus both in its message and in its presentation. Many issues are raised and apparently disregarded. It feels like the paper should have been split up according to its multiple contributions.

As is, many question remain. Most importantly, while the presented approach of MDPs difference appears very good, it is presented without alternatives, which are certainly easy to come by. It seems that the focus on difference between follow-up MDPs naturally implies the properties that are later derived. A comparison of various metrics at this point would have probably made a nice paper on its own.

The convergence analysis of the baseline algorithms seems uninteresting, when they are outperformed in the later study.

The experiments come with a warning regarding their small scale, which should be noted. But at this scale, it is hard to see them as more as just a sanity check on the analytical approach. What happens when the MDPs are more different than the very small variations presented? Can a changed state/action structure be accounted for somehow or is it remaining fixed integral to the approach?

Minor notes:
- p1. "it suggest" without any it being referred to.
- p2. "it is less studied about" makes no sense to me as a sentence.
- p3. Def 3 ends in the middle of a sentence.
- p5. The very prominently marked question is never given a distinct answer.
- p6. "will leads" should read "will lead"
- p7. The clause starting with "Due to" makes no sense to me. Perhaps replace "is" with "being"?
- p9. typo "Finetuen"

**Questions:**

see above

---

### Official Review · Reviewer_c22e · 2023-11-04

**Soundness:** 2 fair
**Presentation:** 1 poor
**Contribution:** 2 fair
**Rating:** 3
**Confidence:** 2

**Summary:**

The authors propose measuring the squared error distance between value functions corresponding to distinct MDPs, aggregated across state action pairs as a way to quantify catastrophic forgetting. In doing so, the authors make specific technical choices for details such as how to define the state aggregation density (e.g. by the target evironment's steady state distribution).

This is then used to motivate a new re-weighting scheme for the Bellman targets weighted across the distinct task boundaries for a proposed continual learning algorithm to tradeoff plasticity and stability. The algorithm seems easy to implement as a practical extension to  single task Q learning variants and the authors show experimental evidence on simple tabular toy domains.

**Strengths:**

The higher level idea of capturing the drift in value functions as a metric of interest for continual learning makes sense, although the significance of the particular technical instantiations proposed by the authors is unclear both theoretically as well as experimentally.

The authors make good assumptions to narrow down the challenges (e.g. assuming known task boundaries) to extract a more scoped out technical formulation.

The attempt to formalize the objective of catastrophic forgetting is commendable and the specific technical chocies made seem original even if the significance is unclear.

**Weaknesses:**

The main contribution is a new algorithm for continual learning which is derived from the the proposed objective in (4), but this itself is not sufficiently justified. Furthermore, this objective itself depends on both the MDPs and the value estimation algorithm used in each individual, which makes it somewhat unclear to draw conclusions about.

The claimed results on the existing upper bound and lower bound algorithms (e.g. "Finetune algorithm") do not seem to add much in the way of technical insights to prior work.

The experimental work is limited to tabular settings and is difficult to draw any conclusions more generally.

The paper unfortunately has several technical issues with respect notation and other miscellaneous typos, etc. See below for some examples and questions to authors:

- Definition 2, the distance defined is a function of an arbitrary state-action weight function. Is the notation meant to be $d^w_2$ instead of $d^2_2$
- Sec 3.1 "s sequence"
- Task index, t inconsistent with index i used later.
- The main equation defining the update rule for the t^th MDP seems to have some errors wrt super/subscript.
- "conduct a function" --> construct a function?
- Question Sec 4.1: "it is" --> "is it...?"
- s/bulding/building

**Questions:**

Authors say "As suggested in Equation (3), the $\Delta^{\pi_T}_{S,T}(M_S)$ is a weighted MDP difference as defined in Definition 2 for the specified choice of weights. But Definition 1 uses the optimal value functions $Q^*$, whereas the definitions 3,4 are for any given estimation algorithm.

An algorithm usually "regarded as a lower bound" sounds a bit confusing. Is that simply saying that it is considered a good baseline algorithm?

---

### Meta-Review · Area_Chair_urtw · 2023-12-06

**Metareview:**

This paper proposes a continual learning algorithm based on reweighing Bellman targets to balance between plasticity and stability in learning. However, the reviewers generally agree that the experiments in the paper are too limited to toy, tabular domains, while the theoretical analysis of the paper appears disjoint and generally not adding much additional insight. Multiple reviewers also pointed out the paper would benefit from a major copy-editing pass. Thus, the paper in its current state, does not seem like a good candidate for publication at ICLR.

**Justification For Why Not Higher Score:**

- The theoretical discussion in the paper is somewhat disjoint from the proposed method, while offering limited insight
- The experiments focus on overly simplistic tabular environments
- The writing can benefit from a substantial copy-editing pass.

**Justification For Why Not Lower Score:**

N/A

---

### Decision · Program_Chairs · 2024-01-16

Reject